# Chronic Obstructive Pulmonary Disease Effect of Nonapnea Sleep Disorder on the Risk of Obesity: A Nationwide Population-Based Case–Control Study

**DOI:** 10.3390/ijerph19074118

**Published:** 2022-03-30

**Authors:** Pi-Ching Yu, Ren-Jei Chung, Yao-Ching Huang, Shi-Hao Huang, Bing-Long Wang, Wu-Chien Chien, Chi-Hsiang Chung, Chien-An Sun, Sung-Sen Yang, Gwo-Jang Wu

**Affiliations:** 1Graduate Institute of Medical Sciences, National Defense Medical Center, Taipei 11490, Taiwan; yupichin1003@gmail.com (P.-C.Y.); gwojang@yahoo.com (G.-J.W.); 2Cardiovascular Intensive Care Unit, Department of Critical Care Medicine, Far-Eastern Memorial Hospital, New Taipei City 10602, Taiwan; 3Department of Chemical Engineering and Biotechnology, National Taipei University of Technology (Taipei Tech), Taipei 10608, Taiwan; rjchung@ntut.edu.tw (R.-J.C.); ph870059@gmail.com (Y.-C.H.); hklu2361@gmail.com (S.-H.H.); 4Department of Medical Research, Tri-Service General Hospital, Taipei 11490, Taiwan; g694810042@gmail.com; 5School of Public Health, National Defense Medical Center, Taipei 11490, Taiwan; billwang1203@gmail.com; 6Graduate Institute of Life Sciences, National Defense Medical Center, Taipei 11490, Taiwan; 7Taiwanese Injury Prevention and Safety Promotion Association (TIPSPA), Taipei 11490, Taiwan; 8Department of Public Health, College of Medicine, Fu-Jen Catholic University, New Taipei City 242062, Taiwan; 040866@mail.fju.edu.tw; 9Big Data Center, College of Medicine, Fu-Jen Catholic University, New Taipei City 242062, Taiwan; 10Department of Nephrology, Tri-Service General Hospital, Taipei 11490, Taiwan; 11Department of Obstetrics and Gynecology, Tri-Service General Hospital, Taipei 11490, Taiwan

**Keywords:** nonapnea sleep disorder, chronic obstructive pulmonary disease, obesity, National Health Insurance Research Database, case–control study

## Abstract

*Objectives*: To investigate whether chronic obstructive pulmonary disease (COPD) affects nonapnea sleep disorder (NASD) on the risk of obesity. *Materials and Methods*: From 1 January 2000 to 31 December 2015, a total of 24,363 patients with obesity from the 2005 Longitudinal Health Insurance Database were identified; 97,452 patients without obesity were also identified from the same database. Multiple logistic regression was used to analyze the previous exposure risk of patients with obesity and NASD. A *p* value of <0.05 was considered significant. *Results*: The risk of developing obesity in patients with COPD is 3.05 times higher than that in patients without COPD. Patients with COPD with NASD had a 1.606-fold higher risk of developing obesity than those without NASD. Patients with obesity were more likely to be exposed to NASD than did those without obesity (adjusted odds ratio, 1.693; 95% confidence interval, 1.575–1.821, *p* < 0.001). Furthermore, the closeness of the exposure period to the index time was positively associated with the severity of obesity, with a dose–response effect. The exposure duration of NASD in patients with obesity was 1.693 times than that in those without obesity. Longer exposure durations were associated with more severe obesity, also with a dose–response effect. *Conclusions*: The COPD effect of NASD increases the subsequent risk of obesity, and the risk of obesity was determined to be significantly higher in patients with NASD in this case–control study. Longer exposure to NASD was associated with a higher likelihood of obesity, also with a dose–response effect.

## 1. Introduction

Obesity is a worldwide epidemic, and its prevalence is increasing in most Western societies and developing countries. At the current growth rate, the global obesity rate for men and women will reach 18% and over 21%, respectively, by 2025 [1]. Furthermore, obesity may gradually cause or aggravate various complications, including type 2 diabetes, hypertension (HTN), dyslipidemia, cardiovascular disease (CVD), nonalcoholic fatty liver disease (NAFLD), reproductive dysfunction, abnormal breathing, and mental illness. It may even increase the risk of certain types of cancer depending on the degree, duration, and distribution of excess body weight and fat tissue [1]. The World Health Organization (WHO) indicated that obesity is a chronic disease and highlighted the health hazards of obesity. The 2016 WHO data revealed that over 1.9 billion adults (aged ≥18 years) are overweight. Among them, over 650 million were obese [2]. The top ten causes of death in Taiwan in 2017 were cancer, heart disease, CVD, diabetes, hypertensive disease, nephritis, renal syndrome, nephropathy, chronic liver disease, and cirrhosis—seven of which are related to obesity [3]. In the latest survey, the National Health Administration of the Ministry of Health and Welfare reported that the rate of obesity among adults (aged ≥18 years) has increased from 38% in 2009 to 43.9% in 2018 [4]. Compared with people with a healthy weight, obese people have more than three times the risk of diabetes, metabolic syndrome, and dyslipidemia and two times the risk of HTN, CVD, knee arthritis, and gout [4].

Sleep disorders (SDs) are a group of diseases characterized by disturbances in the amount, quality, or timing of sleep or sleep-related physiological conditions [5]. In Taiwan, the prevalence of SDs in the general population is 4.2% [6], and approximately 20% of women aged 25–44 years develop SDs [7]. According to the International Classification of Sleep Disorders (American Academy of Sleep Medicine, 2014), SDs are divided into seven categories: insomnia, sleep-related breathing disorders, central disorders of hypersomnolence, circadian rhythm SDs, parasomnias, sleep-related movement disorders, and other SDs [5]. Nonapnea SDs (NASDs) refer to SDs other than sleep apnea. Studies have extensively investigated the association between SDs and chronic diseases. NASD has recently been associated with an increased risk of other comorbidities, including HTN, diabetes mellitus (DM), chronic kidney disease (CKD), CVD, chronic obstructive pulmonary disease (COPD), and stroke [8]. However, the relationship between SDs and obesity may be a critical mediating factor that links SDs with chronic diseases, including CVD, COPD, and DM, in all age groups [9,10].

COPD is a common respiratory disease characterized by persistent expiratory airflow obstruction, which is progressive and accompanied by chronic airway inflammation [11]. SDs is one of the most common symptoms reported by patients with COPD, occurring in approximately 40% of patients in one large study [11]. These patients have problems initiating or maintaining sleep and have mild increases in sleep, decreases in rapid eye movement (REM) sleep, and frequent sleep stage transitions and microarousals. Sleep efficiency is low in most patient populations [12]. SDs may contribute to the nonspecific daytime symptoms of chronic fatigue, lethargy, and impaired overall quality of life described in 50–70% of patients with COPD [12]. Nocturnal symptoms in patients with COPD are often overlooked by physicians and/or not reported by patients themselves [13]. Understanding this connection may help in developing effective therapeutic interventions for SDs and obesity [14,15]. Longitudinal observational studies investigating the relationship between SDs and obesity are limited. Therefore, we hypothesized that COPD affects NASDs on the risk of obesity. We used the National Health Insurance Research Database (NHIRD) of the Ministry of Health and Welfare to investigate whether the COPD effect of NASDs increases the subsequent risk of obesity. 

## 2. Materials and Methods

### 2.1. Data Source

Taiwan’s National Health Insurance launched the single-payer system on 1 March 1995. As of 2017, 99.9% of Taiwan’s population is enrolled in this program. Data for this study were collected from the 2005 Longitudinal Health Insurance Database (LHID2005), which is part of the NHIRD, and 2,000,000 people were randomly selected from the entire population. The National Institutes of Health encrypt all personal information before releasing the LHID2005 to protect the privacy of patients. In the LHID2005, the disease diagnosis code is based on the *International Classification of Diseases, Ninth Revision, Clinical Modification* (ICD-9-CM) criteria [16]. Figure 1 shows the flowchart of the study design (case–control study) from the NHIRD in Taiwan. All methods were performed in accordance with relevant guidelines and regulations. This study was approved by the Ethical Review Board of the Tri-Service General Hospital of the National Defense Medical Center (TSGHIRB No. B-109-39).

### 2.2. Determining Cases and Controls

This study applied a population-based case–control design. A case–control study is used to determine the causes of a health outcome by comparing two different groups: one group of patients with a particular disease or condition as cases and the other group without a certain disease or condition. Populations with a particular disease or condition are controls. Controls and cases should have similar background characteristics, especially characteristics that affect the disease (or condition) in question (e.g., choosing the same age or the same place of residence), which also means that researchers can use it to discover differences in lifestyle and possible causes of disease. Patients diagnosed with obesity (ICD-9-CM code 278) were defined as an obesity case group. The control group consisted of patients without obesity. Patients in both the case and control groups were matched by the index date, sex, and age at a ratio of 1:4.

### 2.3. Identifying SDs, Obesity, and Comorbidities

The risk factor discussed in this study was SDs, which were defined based on at least three outpatient diagnoses from 2000 to 2015, identified using the ICD-9 codes 780.5 (SDs); 780.50 (SDs, not specified); 780.52 (insomnia, not specified); 780.51, 780.53, and 780.57 (sleep apnea syndrome); 307.4 (specific SDs of nonorganic origin); 780.54 (insufficient sleep, unspecified); 780.55 (24 h sleep–wake cycle interruption, unspecified); 780.56 (dysfunction related to the sleep phase or awakening from sleep); 780.58 (dyskinesia related to sleep, unspecified); and 780.59 (SDs, other).

The outcome of obesity was measured in patients diagnosed with the following conditions: overweight, obesity, and other hyperalimentation (ICD-9-CM code 278), overweight and obesity (ICD-9 CM code 278.0), morbid obesity (ICD-9-CM code 278.01), overweight (ICD-9-CM code 278.02), and obesity hypoventilation syndrome (ICD-9-CM code 278.03).

The comorbidities evaluated in this study were DM (ICD-9-CM code 250), HTN (ICD-9-CM code 401–405), hyperlipidemia (ICD-9-CM code 272.4), CAD (ICD-9-CM code 414.01), stroke (ICD-9-CM code 430–438), chronic heart failure (ICD-9-CM code 428.0), COPD (ICD-9-CM code 490–496), CKD (ICD-9-CM code 585), liver cirrhosis (ICD-9-CM code 571.5), tumor (ICD-9-CM code 199), anxiety (ICD-9-CM code 300.00), and depression (ICD-9-CM codes 296.2, 296.3, 300.4, and 311).

### 2.4. Statistical Analysis

Descriptive data are presented as percentages, means, and standard deviations. Chi-square test and *t*-test were used to evaluate the distribution of categorical and continuous variables between cases and controls. The Charlson comorbidity index (CCI) assesses comorbidity level by considering both the number and severity of 19 predefined comorbid conditions. It provides a weighted score of a client’s comorbidities that can be used to predict short- and long-term outcomes, such as function, hospital length of stay, and mortality rates. After controlling the main reasons for admission and severity, survival analysis was used to explore the relationship between comorbidity and death within 1 year, and one category of comorbidity was weighted according to the adjusted relative risk. The 10-year survival of the patient was verified. Table 1 presents the comorbidity categories and weights. If the relative risk was >1.2 and <1.5, the weight was 1; if the relative risk was >1.5 and <2.5, the weight was 2; if the relative risk was >2.5 and <3.5, the weight was 3. The relative risk of type 2 comorbidity was >6 and was given a weight of 6. The comorbidity weights of patients were aggregated. Conditional logistic regression analyses were performed to evaluate the effect of NASD on the risk of obesity after adjusting for age, sex, education, insured premium, comorbidities, CCI, season, location, urbanization level, and level of care. The effect of the first to last NASD exposure before obesity diagnosis on the factors of obesity was examined using conditional logistic regression. All analyses were performed using SPSS version 22 (IBM, Armonk, NY, USA). A *p* value of <0.05 was considered significant.

## 3. Results

### 3.1. Demographic Data

As presented in Table 1, the mean age of 121,815 patients was 44.30 ± 15.64 years, among whom 42.77% were men and 57.23% were women. A total of 24,363 patients with obesity (cases) and 97,452 patients without obesity (controls) were recruited. Patients in the case group had a higher prevalence of COPD and liver cirrhosis than did those in the control group. In the case group, the CCI, season, location, urbanization level, and level of care were significant.

### 3.2. Logistic Regression of Obesity Variables

As presented in Table 2, a significantly higher risk of obesity was observed in the NASDs group than in the control group (adjusted odds ratio (AOR), 1.693; 95% confidence interval (CI), 1.575–1.821). Men had 0.810 times the risk of obesity than did women (AOR, 0.810; 95% CI, 0.785–0.865). Furthermore, patients aged 45–64 and >65 years had a significantly lower risk of obesity than did those aged 20–44 years (AOR, 0.596; 95% CI, 0.574–0.619, and AOR, 0.415; 95% CI, 0.392–0.439, respectively). The risk of obesity in COPD was 3.050 times higher than that in without COPD (AOR, 3.050; 95% CI, 2.838–3.278). The risk of obesity was significantly lower in autumn and winter (AOR, 0.825; 95% CI, 0.791–0.860, and AOR, 0.863; 95% CI, 0.824–0.902, respectively).

### 3.3. Logistic Regression to Stratify the Obesity Factors of the Listed Variables

As presented in Table 3, the risk of obesity was higher in patients with NASDs than in controls (AOR, 1.693; 95% CI, 1.575–1.821). Women with NASDs had 1.926 times the risk of obesity compared with women without NASDs (AOR, 1.926; 95% CI, 1.791–2.071). The risk of obesity with NASDs was significantly higher in patients aged 20–44 years than in controls (AOR, 1.959; 95% CI, 1.822–2.107). Patients with COPD with NASDs had a 1.606-fold higher risk of developing obesity than those without NASDs (AOR, 1.606; 95% CI, 1.494–1.728). The conditional logistic regression analysis results reveal that patients with NASDs had a significantly higher risk of obesity in spring than controls (AOR, 1.976; 95% CI, 1.838–2.125).

### 3.4. Logistic Regression to Analyze Obesity Factors between Different Periods of NASD Exposure

As illustrated in Figure 2, obese patients were more likely to have experienced NASDs than nonobese patients (AOR, 1.693; 95% CI, 1.575–1.821). Furthermore, the closeness of the exposure duration to the time of the study was positively associated with obesity severity in a dose–response manner (NASDs exposure of <1 year, AOR, 2.386; NASDs exposure of ≥1 and <5 years, AOR, 1.725; NASDs exposure of ≥5 years, AOR, 1.422).

As shown in Figure 3, the mean exposure duration of NASDs in patients with obesity was 1.693 times that in patients without obesity (AOR, 1.693; 95% CI, 1.575–1.821). Furthermore, a longer exposure duration was associated with more severe obesity, with a dose–response effect (NASDs exposure of <1 year, AOR, 1.420; NASDs exposure of ≥1 to <5 years, AOR, 2.240; NASDs exposure of ≥5 years, AOR, 2.863).

## 4. Discussion

The results of this study reveal that male patients had a significantly lower risk of obesity than did female patients, which is consistent with the findings of a study conducted by Kanter et al. [17]. The prevalence of overweight and obesity among men and women varies greatly within and between countries, and more women are obese than men overall [17]. Patients aged 45–64 or >65 years had a significantly lower risk of obesity than did patients aged 20–44 years; this may be because of weight loss or weight control in middle-aged and elderly people due to the disease itself. However, unknown factors may affect this result [18]. Furthermore, the risk of obesity was significantly lower in autumn and winter than in spring, according to the result reported by Ma et al. [19]. The total daily intake in spring is higher than in autumn, with a daily difference in the total intake of 222 kcal/day [19]. The risk of NASDs comorbidities, such as HTN, diabetes, CKD, CVD, and stroke, also increased in spring. Lin et al. reported similar results [20]. Although the physiological mechanism of the association between NASDs and obesity remains unclear, we inferred the underlying mechanism from previous studies, which may provide some insights for our observations. Research articles (mainly cross-sectional and observational) have not clarified whether SDs cause obesity or obesity causes SDs [21]. Additional research with larger sample sizes and controlling for confounding factors is warranted [21].

Our findings also show that the risk of developing obesity in patients with COPD is 3.05 times higher than that in patients without COPD. Patients with COPD with NASDs had a 1.606 times higher risk of developing obesity than without NASDs. Although weight loss is one way to treat COPD, previous studies have shown that approximately 65% of patients with COPD are overweight or obese and that obesity in patients with COPD is associated with other several disease sequelae, such as increased symptoms of dyspnea, higher prescription rates of inhaled drugs, and higher use of healthcare resources [22]. Obesity is independently associated with several CVD risk factors, including diabetes, HTN, dyslipidemia, cancer, sleep apnea, and other major CVDs; patients with COPD are also associated with obesity [23]. In addition, sleep disturbance is one of the most common symptoms reported by patients with COPD. In a large study, sleep disturbance occurred in approximately 40% of patients with COPD [11]. These patients with COPD have problems with maintaining sleep and have mild increases in sleep and decreased REM sleep, frequent changes in sleep stages, and microarousals. COPD, which is common in the elderly with respiratory diseases, not only affects a variety of coexisting diseases but also directly affects the patient’s health and family care willingness [24].

Previous studies have pointed to a bidirectional link between poor sleep and COPD severity scores, i.e., COPD symptoms such as cough and dyspnea may be responsible for poor sleep quality; sleep disorder can lead to adverse outcomes associated with COPD [25]. Sleep efficiency is low in most COPD patient groups; sleep disturbance may lead these patients with COPD to describe chronic fatigue, lethargy, and overall impairment of quality of life. In nonspecific daytime symptoms, nocturnal symptoms in patients with COPD are often overlooked by physicians and/or not reported by patients themselves [26]. Sleep has several effects on breathing, including changes in central respiratory control, lung mechanics, and muscle contractions, which do not adversely affect healthy individuals but may lead to hypoxia in patients with COPD, especially during REM sleep [27]. Therefore, based on previous studies and the findings of this study, we speculate that NASDs may cause the development of COPD and that COPD may lead to the occurrence of obesity.

These pathophysiological factors may explain the association between the COPD effect of NASDs and obesity demonstrated in this study. Our research results reveal that the risk of obesity in patients with COPD was 3.05 times that of patients without COPD. Patients with COPD with NASDs had a 1.606 times higher risk of developing obesity than patients without NASDs. The prevalence of NASD among patients with obesity was 1.693 times than that in those without obesity. Furthermore, the NASDs duration and closeness to the time of the study were both positively related to the severity of obesity. Therefore, the relationship between the occurrence and duration of COPD effect of NASDs and obesity warrants consideration.

This study has several limitations. First, the NHIRD does not provide detailed information, such as that related to alcohol consumption, smoking, eating, and physical activity behaviors, which may affect our findings. Second, body mass index (BMI) was not a variable in our study. Third, although this study was carefully designed and controlled for confounding factors, biases may still exist due to unmeasured or unknown confounding factors (e.g., the onset of depression, the stage of obesity at the time of diagnosis, and drugs that may affect the outcome). A prospective cohort study is recommended to evaluate the relationship between NASDs, COPD, and obesity.

## 5. Conclusions

This study revealed that the relationship between the occurrence and duration of the effect of COPD on NASDs and obesity warrants consideration and that patients with obesity experienced more severe NASDs than did those without obesity. Furthermore, the closeness to the time of the study and the exposure duration were both positively related to the severity of obesity, with a dose–response effect. NASDs may be a risk factor for obesity. Healthcare providers should pay close attention to the relationship between NASDs, COPD, and obesity. Future prospective and experimental studies need to be conducted in order to better determine a cause-and-effect relationship between NASDs, COPD, and obesity. The research results will be important evidence for healthcare management when promoting the prevention of SDs and obesity.

## Figures and Tables

**Figure 1 ijerph-19-04118-f001:**
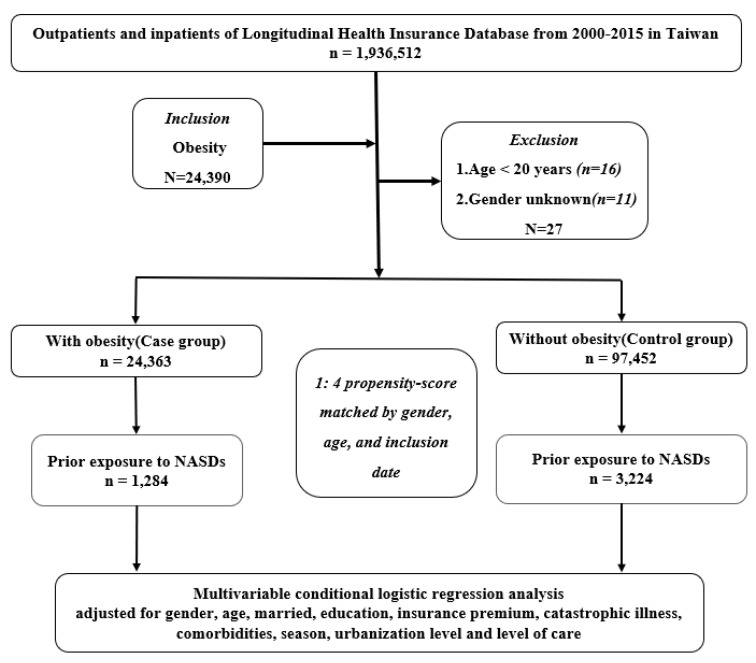
Flowchart the study design from the National Health Insurance Research Database in Taiwan.

**Figure 2 ijerph-19-04118-f002:**
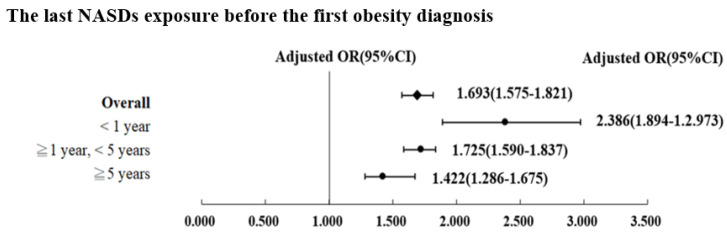
The last nonapnea sleep disorder exposure before the first obesity diagnosis.

**Figure 3 ijerph-19-04118-f003:**
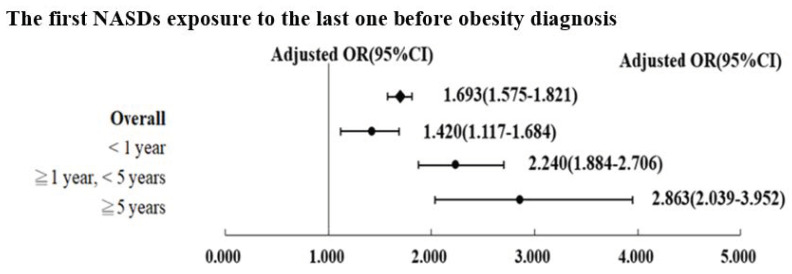
The first nonapnea sleep disorder exposure to the last one before obesity diagnosis.

**Table 1 ijerph-19-04118-t001:** Characteristics of patients.

Obesity	Total		Cases	Controls	*p* Value
Variables	*n* %		*n* %	*n* %	
**Total**	121,815		24,363 20.00	97,452 80.00	<0.001
	NASD				
Without	117,307 96.30		23,079 94.73	94,228 96.69	
With	4508 3.70		1284 5.27	3224 3.31	
	**Sex**				0.999
**Male**	52,105 42.77		10,421 42.77	41,684 42.77	
**Female**	69,710 57.23		13,942 57.23	55,768 57.23	
**Age (mean ± SD, years)**	44.30 ± 15.64		44.25 ± 15.53	44.31 ± 15.67	0.592
	**Age group (years)**				0.999
**20–44**	74,135 47.48		14,827 47.48	59,308 47.48	
**45–64**	34,330 21.99		6866 21.99	27,464 21.99	
**≥65**	47,680 30.54		9536 30.54	38,144 30.54	
	**Married**				<0.001
Without	55,599 45.64		11,298 46.37	44,301 45.46	
With	66,216 54.36		13,065 53.63	53,151 54.54	
	**Education (years)**				
	<12	64,421 52.88	12,897 52.94	51,524 52.87	0.769
	≥12	57,394 47.12	11,466 47.06	45,928 47.13	
	**Insured premium (NT$)**				
	<18,000	119,074 97.75	23,808 97.72	95,266 97.76	0.173
	18,000–34,999	1933 1.59	409 1.68	1524 1.56	
	≥35,000	808 0.66	146 0.60	662 0.68	
	**COPD**				
	Without	118,017 96.88	22,841 93.75	95,176 97.66	<0.001
	With	3798 3.12	1522 6.25	2276 2.34	
	**Liver cirrhosis**				
	Without	121,691 99.90	24,300 99.74	97,391 99.94	<0.001
	With	124 0.10	63 0.26	61 0.06	
	**CHF**				
	Without	121,765 99.96	24,353 99.96	97,412 99.96	<0.001
	With	50 0.04	10 0.04	40 0.04	
**CCI_R**	0.04 ± 0.35		0.05 ± 0.37	0.04 ± 0.35	<0.001
	**Season**				<0.001
Spring (Mar–May)	28,696 23.56		6246 25.64	22,450 23.04	
Summer (Jun–Aug)	30,814 25.30		6513 26.73	24,301 24.94	
Autumn (Sep–Nov)	32,794 26.92		6029 24.75	26,765 27.46	
Winter (Dec–Feb)	29,511 24.23		5575 22.88	23,936 24.56	
	**Location**				<0.001
Northern Taiwan	53,640 44.03		13,152 53.98	40,488 41.55	
Middle Taiwan	32,803 26.93		5118 21.01	27,685 28.41	
Southern Taiwan	27,980 22.97		4793 19.67	23,187 23.79	
Eastern Taiwan	6836 5.61		1250 5.13	5586 5.73	
Outlets islands	556 0.46		50 0.21	506 0.52	
	**Urbanization level**				<0.001
1 (The highest)	39,975 32.82		8823 36.21	31,152 31.97	
2 (Second)	51,737 42.47		9840 40.39	41,897 42.99	
3 (Third)	10,931 8.97		1865 7.66	9066 9.30	
4 (The lowest)	19,172 15.74		3835 15.74	15,337 15.74	
	**Level of care**				<0.001
Hospital center	40,650 33.37		9198 37.75	31,452 32.27	
Regional hospital	56,112 46.06		12,305 50.51	43,807 44.95	
Local hospital	25,053 20.57		2860 11.74	22,193 22.77	

*p*, chi-square/Fisher’s exact test for categorical variables and *t*-test for continuous variables. CHD, coronary heart disease; CHF, chronic heart failure; COPD, chronic obstructive pulmonary disease; CCI, Charlson comorbidity index.

**Table 2 ijerph-19-04118-t002:** Logistic regression of obesity variables.

Variables	Adjusted OR	95% CI	*p* Value
**NASD**			
Without	Reference		
With	1.693	1.575–1.821	<0.001
**Sex**			
Male	0.810	0.785–0.865	<0.001
Female	Reference		
**Age group (years)**			
20–44	Reference		
45–64	0.596	0.574–0.619	<0.001
≥65	0.415	0.392–0.439	<0.001
**Married**			
**Without**	0.872	0.411–1.298	0.583
**With**	Reference		
**Education (years)**			
**<12**	Reference		
**≥12**	0.831	0.311–1.385	0.682
**Insured premium (TWD)**			
**<18,000**	**Reference**		
**18,000–34,999**	1.051	0.935–1.182	0.401
**≥35,000**	0.895	0.740–1.081	0.249
**COPD**			
Without	Reference		
With	3.050	2.838–3.278	<0.001
**Liver cirrhosis**			
Without	Reference		
With	0.000		<0.999
**CHF**			
Without	Reference		
With	0.000		<0.999
**Season**			
Spring	Reference		
Summer	0.990	0.950–1.032	0.651
Autumn	0.825	0.791–0.860	<0.001
Winter	0.863	0.824–0.902	<0.001
**Urbanization level**			
**1 (highest)**	1.853	1.812–1.897	<0.001
**2 (second)**	1.696	1.655–1.729	<0.001
**3 (third)**	1.659	1.616–1.704	<0.001
**4 (lowest)**	Reference		
**Level of care**			
**Hospital center**	2.593	2.461–2.731	<0.001
**Regional hospital**	2.269	2.162–2.370	<0.001
**Local hospital**	Reference		

Adjusted OR, adjusted odds ratio; CI, confidence interval; CHF, chronic heart failure; COPD, chronic obstructive pulmonary disease.

**Table 3 ijerph-19-04118-t003:** Factors of obesity stratified by variables using logistic regression.

Group	With NASD vs. without NASD *(Reference)*		
Stratified	Adjusted OR	95% CI	*p* Value
**Total**	1.693	1.575–1.821	<0.001
**Sex**			
Male	1.408	1.310–1.515	<0.001
Female	1.926	1.791–2.071	<0.001
**Age group (years)**			
20–44	1.959	1.822–2.107	<0.001
45–64	1.545	1.438–1.662	<0.001
≥65	1.392	1.295–1.498	<0.001
**Married**			
Without	1.583	1.472–1.702	<0.001
With	1.777	1.653–1.911	<0.001
**Education (years)**			
**<12**	1.702	1.584–1.831	<0.001
**≥12**	1.683	1.565–1.810	<0.001
**Insured premium (TWD)**			
**<18,000**	1.509	1.403–1.623	<0.001
**18,000–34,999**	1.700	1.581–1.828	<0.001
**≥35,000**	1.156	1.076–1.244	0.002
**COPD**			
Without	1.598	1.486–1.718	<0.001
With	1.606	1.494–1.728	<0.001
**Liver cirrhosis**			
Without	1.693	1.575–1.821	<0.001
With			
**CHF**			
Without	1.693	1.575–1.821	<0.001
With			
**Season**			
Spring	1.976	1.838–2.125	<0.001
Summer	1.767	1.644–1.901	<0.001
Autumn	1.461	1.360–1.572	<0.001
Winter	1.722	1.602–1.852	<0.001
**Urbanization level**			
1 (highest)	1.874	1.743–2.016	<0.001
2 (second)	1.768	1.644–1.901	<0.001
3 (third)	1.591	1.480–1.711	<0.001
4 (lowest)	1.412	1.314–1.519	<0.001
**Level of care**			
Hospital center	2.451	2.280–2.636	<0.001
Regional hospital	1.414	1.315–1.521	<0.001
Local hospital	1.387	1.290–1.491	<0.001

Adjusted OR, adjusted odds ratio (adjusted for the variables listed in Table 3); CI, confidence interval; CHF, chronic heart failure; COPD, chronic obstructive pulmonary disease.

## Data Availability

Data are available from the NHIRD published by the Taiwan National Health Insurance Administration. Due to legal restrictions imposed by the government of Taiwan concerning the Personal Information Protection Act, data cannot be made publicly available. Requests for data can be sent as a formal proposal to the NHIRD (http://www.mohw.gov.tw) (accessed on 13 January 2022).

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
