# Peer review of "Chronic Obstructive Pulmonary Disease Effect of Nonapnea Sleep Disorder on the Risk of Obesity: A Nationwide Population-Based Case–Control Study"

_ijerph, 2022, doi:10.3390/ijerph19074118_

Round 1

Reviewer 1 Report

Though the causality remains to be addressed in a cross sectional study, the findings would be of value. It may be difficult to conclude that COPD affects the incidence of NASD or obesity. Precisely, this study can address the incidence of the disorders such as obesity or NASD in COPD or non-COPD because the onset of NASD and COPD are not investigated in the cross sectional study. So please avoid the overestimation from the findings saying that COPD increases the incidence of obesity or NASD. The link between COPD, obesity and NASD would be of value to be fully described, however, the confounding factors should be explained in the discussion as the authors did, and please confirm that the discussion is based on the previous reports as the authors referred in the response.  The expression would be fully verified based on the findings in the cross sectional study.

Reviewer 2 Report

The authors responded and solved the issues from the first manuscript. 

Reviewer 3 Report

The revised version of the manuscript is improved.

Author Response

See attachment file.

This manuscript is a resubmission of an earlier submission. The following is a list of the peer review reports and author responses from that submission.

Round 1

Reviewer 1 Report

This epidemiological study aimed to examine the association among obesity, NASD, and COPD.

There are several significant concerns.

  1. Could the cross sectional epidemiological study address the causations of the disorders?
  2.  Generally, patients with COPD tend to be lower BMI than controls. In Taiwan, does that  tendency  exist?
  3. COPD is characterized by airflow limitation and the pathophysiology of COPD consists of small airway disease and emphysema. How does NASD affect the development of COPD?
  4. What is the mechanism of COPD to lead to obesity?
  5. Could it be possible that obesity is the basic cause to lead to NASD? Is smoking rate higher in obesity patients compared with cases without obesity? If so, many confounders such as inappropriate diet and less exercise, resulting in many disorders may exist.
  6.  Cross sectional studies may not address the causality of the diseases, therefore, the massages should be carefully re-considered. 
  7. Is the incidence of diseases possibly examined as the longitudinal study? If so, the causality might be more clear than in the current analysis.
  8. The changes in data presentation could make the massages more properly understood. For instance, the visually informative figures or dichotomous tables may help the understandings of the readers.     

Reviewer 2 Report

The paper "Chronic Obstructive Pulmonary Disease effect of Nonapnea 
Sleep Disorder on the Risk of Obesity—A Nationwide Population-Based Case Control Study by Pi-Ching Yu and colleagues presents an interested approach of a common association of diseases, with a solid statistical analysis and an expected result. There are some issues that must be solved.

  1. Page 2. There are repeated abbreviation of comorbidities (COPD, CVD) and I think you need to be consisted.
  2. Page 2. Reference 5 is inaccurate. It does not referred to AASM 2014 classification you mentioned in the text.
  3. Page 3.  Figure 1. You need a better quality. 
  4. Page 3. Determining cases and controls. Can you add some information about the particularities of obesity in Asia? Including  a stratification based on BMI?  
  5. Page 4. CKD explanation, CCI definition. Demographic data: mean instead of average. 
  6. Page 7. Figure 2 needs a better quality. 
  7. Page 2. Repeated description of comorbidities that are already presented with abbreviation. It is a bit confusing. 
  8. Page 8. Reference 26 is similar with the reference 13. 
  9. Page 9. Limitations. A strong point. 
  10. Page 10. References. The most recent one is a report from 2020. 13 is the same with 26.  

Reviewer 3 Report

Manuscript ID: ijerph-1592645

Title: "Chronic obstructive pulmonary disease effect of nonapnea sleep disorder on the risk of obesity – a national population-based case control study"

The aim of this study was to investigate the possible role of Chronic Obstructive Pulmonary Disease (COPD) effect of Nonapnea Sleep Disorder (NASDs) on the risk of obesity. Data analysed derived from National Health Insurance Research Database (NHIRD) of the Ministry of Health and Welfare. Overall, results support the hypothesis put forward by Authors.

The topic of the manuscript is potentially interesting.

The drafting of the manuscript is sufficiently clear.

The authors seem quite aware of the inherent limitations of the research, which in fact are clearly explained in the discussion section.

I would suggest adding more informative legends for both the tables and the figures.

It seems to me that the quality of the figures should be improved.

Considering the limitations and the correlational nature of the analyses, the discussion section could be a bit shorter without commenting data that have not been analysed in the present research.